# Performance Evaluation of the Two-Input Buck Converter as a Visible Light Communication High-Brightness LED Driver Based on Split Power

**DOI:** 10.3390/s24196392

**Published:** 2024-10-02

**Authors:** Daniel G. Aller, Diego G. Lamar, Juan R. García-Mere, Manuel Arias, Juan Rodriguez, Javier Sebastian

**Affiliations:** 1Airbus Crisa (an Airbus Defense and Space Company), 28760 Madrid, Spain; daniel.garcia-aller@airbus.com; 2Power Supplies Group, Electrical Engineering Department, University of Oviedo, 33204 Gijon, Spain; gonzalezdiego@uniovi.es (D.G.L.); garciamjuan@uniovi.es (J.R.G.-M.); ariasmanuel@uniovi.es (M.A.); sebas@uniovi.es (J.S.)

**Keywords:** visible light communication (VLC), high-brightness LED, two-input buck (TIBuck) DC/DC converter, split power

## Abstract

This work proposes a high-efficiency High-Brightness LED (HB-LED) driver for Visible Light Communication (VLC) based on a Two-Input Buck (TIBuck) DC/DC converter. This solution not only outperforms previous approaches based on Buck DC/DC converters, but also simplifies previous proposals for VLC drivers that use the split power technique with two DC/DC converters: one is in charge of the communication tasks and the other controls the biasing of the HB-LED (i.e., lighting tasks). The real implementation of this scheme requires either two input voltage sources, one of which is isolated, or one DC/DC converter with galvanic isolation. The proposed implementation of splitting the power is based on a TIBuck DC/DC converter that avoids the isolation requirement, overcoming the major drawback of this technique, keeping high-efficiency and high communication capability thanks to the lower voltage stress both across the switches and at the switching node. This fact allows for the operation at very high frequency for communication purposes, minimizing switching power losses, achieving high efficiency and providing lower filtering effort. Moreover, the duty ratio range can also be adapted to the useful voltage range of the HB-LED load to maximize the resolution on the tracking of the output volage. The power is split by means of an auxiliary Buck DC/DC converter operating at low switching frequency, which generates the secondary voltage source needed by the TIBuck DC/DC converter. This defines a natural split of power by only processing the power delivered for communications purposes at high frequency. A 7 W output-power experimental prototype of the proposed VLC driver was built and tested. Based on the experimental results, the prototype achieved 94% efficiency, reproducing a 64-QAM digital modulation scheme and achieving a bit rate of 1.5 Mbps with error in communication of 12%.

## 1. Introduction 

Most wireless protocols and technologies are currently based on the use of the Radio Frequency (RF) spectrum, such as 5G, WiFi, Bluetooth, etc. The current overuse of the spectrum and projected growth of users and data rates [1,2] have led to an increasing interest on finding new wireless technologies to either complement or replace current wireless networks. One of the most promising alternatives in the recent years has been Visible Light Communication (VLC) technology [3,4,5,6].

VLC uses the visible light spectrum (from 380 nm to 780 nm) and takes advantage of the widespread use of High-Brightness LEDs (HB-LEDs) in Solid-State Lighting (SSL). The capability of HB-LEDs to rapidly change the light emitted makes them suitable for incorporating communication capability in SSL systems.

To use HB-LEDs in VLC systems, the VLC driver needs to do two things: the bias tasks (i.e., lighting) and the communication tasks. In a traditional lighting application based on HB-LEDs, the driver needs to perform the bias task by maintaining a desired Q-point (i.e., bias point). This means that the HB-LED driver controls the average illumination level and counteracts any temperature-related effects on the HB-LEDs. The temperature behaviour of the HB-LED and its control is illustrated in Figure 1. This figure shows the effect of temperature on the HB-LED in the I-V-Flux characteristic at two different temperatures, T_1_ and T_2_, where T_2_ is greater than T_1_. The most notable effect is observed in the threshold voltage, making the V-I characteristic of the HB-LED shift to lower voltages as the temperature rises. Even though the temperature also affects the I-Flux characteristic, the effects are negligible and have less impact on communication performance than the threshold voltage shift [7,8,9].

The voltage applied to the HB-LED is v_o_(t) Its average value, V_bias_(T), must be controlled to keep the average value of io(t), I_bias_, constant and controlled. Note that the temperature dependency is shown by the (T) at the end of the names. The bias task is normally performed by means of a feedback loop implemented throughout the HB-LED driver. Lighting-level control (i.e., dimming) and temperature-effect mitigation are also expected in VLC applications, necessitating average output current control. The communication tasks are what make the traditional HB-LED driver into a VLC driver. To implement communication capability, a high frequency communication signal must be applied around the Q-point. This is also depicted in Figure 1, with an example of a communication.

To minimize the distortion, the HB-LED must work in its linear region, avoiding the vicinity of the threshold voltage V_th_(T). Another limit is the maximum current of the HB-LED, I_max_. This current defines a maximum voltage V_max_(T), which depends on the temperature, as shown in Figure 1. Therefore, the working voltage region of the HB-LED, as a load of the VLC driver is derived: V_Ω_ = V_max_(T) − V_th_(T). It is worth noting that V_Ω_ is not temperature-dependent, which this work will take full advantage of.

The peak-to-peak amplitude of the communication signal, V_pp_, and its average value, V_bias_(T), are selected following a certain criteria. In this case, the criterion is maximizing the communication signal amplitude, making
(1)$$V_{pp}=V_{\Omega },$$

This need means that the Q-point is fixed in the middle of the linear region, leading to
(2)$$V_{bias}\left(T\right)=V_{th}\left(T\right)+\frac{V_{\Omega }}{2},$$
and
(3)$$I_{bias}=\frac{I_{max}}{2}.$$

It is important to note that the maximization of the communication signal amplitude (i.e., to maximize the communication distance range) constrains the value of the Q-point, (i.e., the value of I_bias_). This imposes a limit on the average light emitted by the HB-LED. Additionally, since V_Ω_ does not depend on temperature, it means that the maximum peak-to-peak value of signal V_pp_ is also constant.

The addition of the communication signal has some implications for the HB-LED that are worth mentioning. Its linear behaviour plays an important role in communication tasks. Some types of pre- and post-equalization have been proposed to counteract non-linearities, making the HB-LED able to reach higher peak-to-peak values [10,11,12], but those depend on the modulation scheme used and they are applied directly on the generation of the communication signal, with no impact on VLC driver design.

Another important consideration is the dimming capability of the VLC driver. The illumination level is controlled by the average current, I_bias_. But, if the communication signal is maximized, this means that I_bias_ will have a fixed value and cannot be modified (i.e., no dimming). In a VLC driver, the dimming capability is strongly linked to the communication modulation scheme and these must all be implemented together [5,6,13]. The criteria chosen in this work is a worst-case scenario in terms of efficiency. Maximizing the communication signal means that the power processed at high frequency is also maximized, making the design more challenging in terms of reducing switching-power losses, but meaning that the communication covers a greater distance.

HB-LEDs also impose a limitation on the bandwidth of the VLC driver. In general, a blue gallium nitride HB-LED in combination with a yellow inorganic phosphor is the preferred approach for obtaining white light in SSL. However, this phosphor limits the HB-LED bandwidth to a few MHz (3–5 MHz) [14,15].

All these constraints for HB-LEDs must be considered in the design of VLC drivers. Most VLC driver topologies are based on the use of regular HB-LED drivers for the bias tasks connected in parallel to a bias T scheme with an RF linear power amplifier (i.e., class A or B) [16,17]. This solution leads to simple implementations with high bit rates. The main disadvantage is decreased efficiency due to using RF linear power amplifiers with theoretical maximum efficiencies of 50% and 78%, respectively, for class A and B. Moreover, efficiencies fall sharply when high bit rates and more complex modulation schemes are used (between 10% and 40%). The main advantage of RF linear power amplifiers is the wider bandwidth they can achieve, but this advantage cannot be exploited with current HB-LED lighting technology.

Another approach is based on merging the two tasks, bias and communication, into the HB-LED driver, making a DC/DC converter based on conventional Pulse-Width Modulation (PWM), also able to generate the communication signal, but, in this case additional signal distortion is introduced by the switching frequency. The capability of the fast-response and high-frequency DC/DC converters to track their output voltage according to a communication signal at high efficiency has previously been reported and exploited in communication applications such as Envelope Tracking (ET) and Envelope Elimination and Restoration (EET) [18,19,20]. These DC/DC converters can achieve bandwidths around tens of MHz with high efficiency. They have been proposed as VLC drivers, achieving efficiencies around 90% and high bit rates [21]. One of the disadvantages of using the same DC/DC converter for both biasing and communication tasks is that the bias power is processed at high switching frequency. In a VLC driver, the biasing power is much higher than the communication power, and there is no need to use a high-frequency converter for the biasing process.

This context is where the technique of splitting the power between two DC/DC converters is proposed, to further improve the efficiency of the VLC driver [22]. As Figure 2a shows, this technique is based on using two converters: a low-switching-frequency DC/DC converter, acting as an HB-LED driver and the other high-switching-frequency and fast-response DC/DC converter, providing the communication tasks. In this case additional signal distortion is introduced by the switching frequency, too. This configuration leads to a partial power conversion between the two converters, where the power is divided depending on the task. The architecture proposed in [22] uses a regular Buck DC/DC converter with low switching frequency operation for lighting tasks and a two-phase Buck DC/DC converter with high-order output filter, operating at high switching frequency for communication tasks. The main drawback is the need for two input voltage sources, one of which is isolated, for the real implementation, adding hardware complexity to the design of the previous stages of the power supply chain. Following this technique other approaches could be taken, as shown in Figure 2b, using isolated DC/DC converters. However, any configuration based on an isolated DC/DC converter for splitting power introduces the same hardware complexity as mentioned above, but in the last step of the power supply chain: the VLC driver.

The objective of this work is the proposal of a new architecture of a VLC driver based on the split power technique. The idea is to use the same partial power conversion between two DC/DC converters to achieve high efficiency in the VLC driver, improving on the performance of traditional Buck DC/DC converter solutions and overcoming the hardware complexity of traditional schemes based on split power [22]. To take advantage of the integration between VLC and SSL infrastructure, this work focuses on VLC drivers for HB-LEDs, meaning that the maximum bandwidth needed is limited to 5 MHz.

This paper introduces a high-efficiency HB-LED driver for VLC based on the Two-Input Buck (TIBuck) DC/DC converter. The proposed TIBuck DC/DC converter only needs one input voltage source to naturally split the power with high efficiency and high communication capability, overcoming the weakness of previous approaches because no galvanic isolation is needed and, therefore, the hardware complexity is simplified. Therefore, the main advantages for splitting the power are reached naturally. The lower switching harmonic components and higher duty cycle resolution of the TIBuck DC/DC converter allows good communication performance from the VLC driver. Moreover, the lower voltage stress across switches achieves high efficiency. Power is split by means of an auxiliary Buck DC/DC converter, which is in charge of generating the secondary voltage source needed by the TIBuck DC/DC converter. This gives the division of the tasks regarding power conversion. Due to the separation of tasks, each converter has different requirements, allowing further optimization of each, according to its task.

The paper is organized as follows. Section 2 reviews the operating principles of the TIBuck DC/DC converter, highlighting its main advantages over Buck DC/DC converters for use as a VLC driver. Section 3 describes the proposed VLC driver based on split power with only one input voltage source, using both an auxiliary Buck DC/DC converter and a TIBuck DC/DC converter. Section 4 covers a theoretical efficiency study, which is needed both to identify the amount of the power for switching and the amount of power that comes up to the load directly. This identification means that the power conversion made at different switching frequencies can be quantified. The objective is to maximize efficiency by keeping the communication features of the VLC driver (i.e., communication capability and resolution) at a maximum. The experimental results are given in Section 5, and finally, the conclusions are described in Section 6.

## 2. Working Principle of the TIBuck DC/DC Converter

### 2.1. Formatting of Mathematical Components

The TIBuck DC/DC converter was originally proposed as the last stage of the power supply chain of two-stage AC/DC single-phase converters [23], because it presents lower output ripple noise and higher efficiency, due to the lower voltage stress and its partial power conversion in comparison to the Buck DC/DC converter. Due to its features, in subsequent work, the TIBuck DC/DC converter has been proposed for different applications, such as the first-stage in photovoltaic power conversion [24]. The potential use of the TIBuck DC/DC converter as a VLC driver was mentioned in [21].

### 2.2. Working Principle

Figure 3 shows a representation of the TIBuck DC/DC converter and its control. This DC/DC converter is a modification of the traditional Buck DC/DC converter, in which the diode S_ti_ is connected to an auxiliary input voltage, V_2_, instead of the negative terminal of the main input voltage, V_1_. For proper operation, V_1_ and V_2_ must comply with the following inequation
(4)$$V_{1}>V_{2},$$
where the traditional Buck DC/DC converter is a particular case when V_2_ = 0 V. Due to the addition of V_2_, the switching node voltage v_sw_(t) varies between V_1_ and V_2_, as shown in Figure 4. The PWM signal v_gs−ti_(t) that controls the main switch Q_ti_ is obtained by comparing a sawtooth waveform with the reference signal. This reference is traditionally considered constant in DC/DC converters, whereas it varies in VLC drivers based on DC/DC converters. For the sake of simplicity, in this case the reference is depicted as a sinusoidal waveform, but it could be more complex. The comparison between the sawtooth signal and the reference modulates the duty ratio d_ti_(t) and consequently, v_gs−ti_(t) (see Figure 4).

In high-switching-frequency converters, this modulation could be implemented in a digital platform, called Digital PWM (DPWM). Analog implementations of high-frequency PWM are also possible, but integration with the communication task makes them unfeasible in practice for VLC drivers. The DPWM consists of a digital counter and a high-frequency clock, as depicted in Figure 3. For the sake of representation, Figure 5 shows an enlargement of two switching periods. From a high-frequency clock CLK, whose frequency is f_clk_, the DPWM counts each clock cycle, T_clk_, until a certain C_sw_ value. The frequency of the sawtooth signal is then
(5)$$f_{sw}=\frac{f_{clk}}{C_{sw}},$$
which will be the switching frequency of the converter.

In each clock period (i.e., T_clk_), the reference signal and the sawtooth are compared. Due to the discretization of the process, there are only C_sw_ possible values for the duty cycle d_ti_(t), which impacts the resolution of the DC/DC converter for tracking the output voltage.

The output voltage of the DPWM, v_gs−ti_(t), controls the MOSFET Q_ti_ of the TIBuck DC/DC converter through its driver. The voltages V_1_ and V_2_, together with Q_ti_ and the diode S_ti_, make up a switching node whose output voltage is v_sw_(t), also shown in Figure 4. This square waveform is applied to the input of the low-pass output filter, producing the output voltage v_o_(t). The output voltage would follow the reference signal if, and only if, certain conditions regarding the frequencies involved and the filter design are met. These conditions have been deduced for the Buck DC/DC converter [18] and are applicable to the TIBuck DC/DC converter.

With f_sig_ the centre frequency of the reference communication signal spectrum, and f_sig−max_ its maximum frequency, f_c_ the cutoff frequency of the filter and f_sw_ the switching frequency of the PWM signal, the frequency conditions can be written as
(6)$$f_{sig-max}>\frac{f_{sw}}{2},$$
(7)$$f_{sig-max}<f_{c}<f_{sw}.$$

Equation (6) is imposed by the Nyquist–Shannon sampling theorem, since the comparison between the sawtooth and the reference works as a sampler. Equation (7) comes from the filtering action, allowing the filter to separate the communication signal frequencies from the switching harmonics. Equation (7) is effectively the most restrictive, because if Equation (7) is met, Equation (6) is also met. This relation between frequencies defines the trade-off between the bandwidth of the VLC driver, the switching frequency, and the switching harmonic components at the output. The left-hand side of Equation (7) defines the bandwidth, restricting the maximum frequency of the signal. The right-hand side restricts the lower switching frequency.

If the latter inequations are fulfilled, (6), (7), and the output voltage, v_o_(t), would only have the DC component plus the spectrum of the communication signal to be transmitted. Then, v_o_(t) can be written in terms of the input voltages V_1_ and V_2_, and the duty cycle d_ti_(t), as follows
(8)$$v_{o}\left(t\right)=v_{sig}\left(t\right)+V_{2}={\left(V_{1}-V_{2}\right)\cdot d_{ti}\left(t\right)+V}_{2}.$$

The communication signal v_sig_(t) is defined as the variation of v_o_(t) due to d_ti_(t). Equation (8) helps us to deduce the key characteristics of the TIBuck DC/DC converter. The output voltage is controlled by the duty cycle d_ti_(t) and varies from V_1_ (when d_ti_(t) becomes 1) and V_2_ (when d_ti_(t) becomes 0), as depicted in Figure 4.

### 2.3. Advantages over the Buck DC/DC Converter

As introduced in [21], the TIBuck DC/DC converter outperforms the traditional Buck DC/DC converter in terms of lower harmonic switching components, higher resolution to track the output voltage and lower switching losses.

Lower switching harmonic components. The lower the amplitude of the switching node voltage, v_sw_(t), the lower the switching harmonic components. These harmonic components result in undesirable output voltage noise and must be attenuated by the output filter. Reducing the harmonic components of v_sw_(t) simplifies the design of the output filter compared to the equivalent in a traditional Buck DC/DC converter, allowing either the reduction in filter order (reducing hardware complexity and number of elements) or increased cut-off frequency (increasing the bandwidth of the converter) [19]. Figure 6 depicts the filtering process, illustrating the separation between frequencies stated by Equation (7).Lower switching losses. The switching losses mostly depend on the switching frequency, characteristics of the switches and voltage stress [25]. According to Equation (7), for a given bandwidth, the switching frequency must be kept high enough for the correct operation of the DC/DC converter as a VLC driver. Increasing the switching frequency of the DC/DC converter leads to a significant increase in switching losses. To track communication signals into the bandwidth of HB-LEDs, the switching frequency of the VLC driver would need to be in the range of 10–15 MHz. The characteristics of the switches depend on the technology available, and it is assumed that the best possible switch could always be used. At this point, only the voltage stress of the switches can be modified to reduce the switching losses In a TIBuck DC/DC converter, the voltage stress of the switch is V_1_ − V_2_, which is lower than the regular Buck DC/DC converter case (i.e., V_1_). This reduction in the voltage stress is also an advantage in switch selection, because the lower the voltage rated, the better the characteristics and performance at high-switching-frequency operation.Higher duty-cycle resolution. Increased switching frequency has a negative effect on the duty-cycle resolution (i.e., number of discrete values of the duty cycle, C_sw_, using a digital control). From Equation (5), for a fixed clock frequency f_clk_, if f_sw_ increases, then the resolution of the duty cycle decreases (i.e., by the C_sw_ decrease). On top of this decrease in the duty-cycle resolution in high-frequency converters, the behaviour of the HB-LED as a load imposes a further decrease in the resolution, which is the case in designing VLC drivers. As depicted previously in Figure 1, only the useful voltage range between V_th_(T) and V_max_(T) can be used for HB-LED as a load, to avoid distortions in the safe operating range (in this initial approach, temperature dependency is ignored). This means that only the values of the duty cycle that generate voltage levels in this range are useful. Figure 7 represents the useful duty-cycle range of a Buck DC/DC converter and a TIBuck DC/DC converter (optimized for this task) using a string of n HB-LEDs as a load (for the sake of simplification, sinusoidal waveforms are used as communication signals). The duty-cycle range of a Buck DC/DC converter can be easily obtained from Equation (8) just by making V_2_ = 0 V. Its output voltage range goes from 0 V to V_1_ = nV_max_(T) by maximizing the communications signal (i.e., V_Ω_ = V_pp_). That means that there is no way to fit the C_sw_ possible duty-cycle values within the useful voltage range of the HB-LED load, which further reduces the resolution for tracking the output voltage (i.e., higher step height in the digital sawtooth, in green, in Figure 7). However, the output voltage range of the TIBuck DC/DC converter goes from V_2_ to V_1_, and can be adapted to fit all C_sw_ possible duty-cycle values within the useful voltage range of the HB-LED load, maximizing the resolution of the TIBuck DC/DC converter for tracking the output voltage (i.e., lower step height in the digital sawtooth, in red, in Figure 7). For this purpose, the following expressions must be met.


(9)
$$V_{1}=n\cdot V_{max}\left(T\right),$$



(10)
$$V_{2}=n\cdot V_{th}\left(T\right).$$


As a summary of this section, reducing harmonic switching components, lower switching losses and the higher duty-cycle resolution make the TIBuck DC/DC converter a suitable option for fast-response and high-frequency applications, such as VLC driving. The following section presents the adaptation of the TIBuck DC/DC converter to work as a VLC driver, naturally splitting the power with only one input voltage source. Moreover, the control strategy of this new solution is addressed.

## 3. The Use of the TIBuck DC/DC Converter as a VLC Driver

### 3.1. The Proposed TIBuck DC/DC Converter with Only One Input Voltage Source

The proposed version of the TIBuck DC/DC converter with only one input voltage source is shown in Figure 8. For the analysis, two individual converters can be considered: the TIBuck DC/DC converter and the auxiliary Buck DC/DC converter. The V_2_ for the TIBuck DC/DC converter can be obtained from V_1_ with the simple addition of an auxiliary Buck DC/DC converter. This auxiliary buck converter does not need either high switching frequency or fast output voltage response, achieving high efficiency.

With d_bu_ the duty cycle of the auxiliary Buck DC/DC converter, voltage V_2_ can be written as
(11)$$V_{2}=d_{bu}\cdot V_{1}.$$

Substituting Equation (11) into Equation (8), the output voltage, v_o_(t), can be written in terms of the duty cycle of both DC/DC converters (TIBuck and auxiliary Buck) as
(12)$$v_{o}\left(t\right)={\left(V_{1}-V_{2}\right)\cdot d_{ti}\left(t\right)+d_{bu}\cdot V}_{1}.$$

As mentioned previously, the duty cycle of the TIBuck DC/DC converter d_ti_(t) comes from modulating a communication signal with the DPWM. The communication signal has an average value and a time variant value, so d_ti_(t) also has them. Therefore, d_ti_(t) can then be written in terms of its average value D_ti_ and the variations ∆d_ti_(t)
(13)$$d_{ti}\left(t\right)=D_{ti}+{\Delta d}_{ti}\left(t\right).$$

As highlighted previously in Figure 7, to maximize both the amplitude of the communication signal and the resolution for tracking the output voltage, using the whole d_ti_(t) range, D_ti_ must be equal to 0.5, meaning that ∆d_ti_(t) can vary from −0.5 to 0.5.

If the switching frequency of the auxiliary Buck DC/DC converter is at least one order of magnitude lower than that used in the TIBuck DC/DC converter, during a whole switching period of the auxiliary Buck DC/DC converter, the effective value of d_ti_(t) can be replaced by its average value D_ti_. This quasistatic condition simplifies the calculation of the average output voltage V_o_, which can be deduced from Equation (12) as
(14)$$V_{o}=avg\left(v_{o}\left(t\right)\right)=0.5\cdot \left(1+d_{bu}\right)\cdot V_{1}.$$

According to Equation (14), the control of the average output voltage, Vo, can be implemented throughout d_bu_. This means that the average current driven by the HB-LEDs, I_bias_, and the average emitted light, Φ_avg_, (depicted in Figure 1) only depends on the auxiliary Buck DC/DC converter control. Therefore, to achieve good performance over temperature changes, the auxiliary Buck converter DC/DC converter would work in a closed loop. On the other hand, the TIBuck DC/DC converter could work in an open loop, simplifying the control design and maximizing bandwidth. Finally, it is important to note that during the switching period of the TIBuck DC/DC converter, the voltage V_2_ can be considered constant. This means that the instantaneous value of the output voltage v_o_(t) is controlled throughout d_ti_(t), according to Equation (12).

### 3.2. Effect of HB-LED Temperature Dependency

Figure 7 and Equations (9) and (10) show the values of V_1_ and V_2_ that maximize resolution for tracking the output voltage of the TIBuck DC/DC converter acting as a VLC driver, which uses the maximum amplitude of the communication signal. These values depend on the threshold voltage nV_th_(T) and the maximum allowed voltage nV_max_(T) of the HB-LED load. But, as mentioned and shown in Figure 1, these values depend on the temperature. Taking this into consideration, Figure 9 shows the effect of temperature changes on the resolution for tracking the output voltage (for sake of simplification, sinusoidal waveforms are used as communication signals). This figure represents the I-V characteristic curve of the HB-LED load for two different temperatures T_1_ (in orange) and T_2_ (in green), where T_1_ > T_2_. The design is optimized for the lower temperature T2, meaning that V_1_(T_2_) and V_2_(T_2_) are obtained using Equations (9) and (10) as follows
(15)$$V_{1}\left(T_{2}\right)=n\cdot V_{max}\left(T_{2}\right),$$
(16)$$V_{2}\left(T_{2}\right)=n\cdot V_{th}\left(T_{2}\right).$$

If the temperature of the HB-LED load rises to T_1_, the voltages V_max_(T) and V_th_(T), which maximize the resolution of the duty cycle, also change. Even though V_2_ can be controlled though d_bu_, according to Equation (14), and is able to match the tracking of the communication signal allowing its maximum amplitude, there is no control over V_1_ (V_1_ temperature changes could only come from control changes of previous stages of the power supply chain). This means a loss of the useful duty-cycle range of the TIBuck DC/DC converter (in blue in Figure 9), and, therefore, a reduction in the resolution for tracking the output voltage. Moreover, the higher the temperature rise of the HB-LEDs, the lower the resolution. This is the price to pay for the simplification compared to other solutions that split power with two DC/DC converters (Figure 2).

The temperature also has a negative effect on switching losses. As previously mentioned, the most significant power losses of the TIBuck DC/DC converter, acting as a VLC driver, are the switching losses due to its high switching frequency. The switching losses depend on the voltage stress of switches (i.e., V1-V2). While V1 is kept constant and V2 decreases with the rise in the temperature, the switch voltage stress increases when the temperature of the HB-LEDs increases. Therefore, the switching losses increase when the temperature rises. This drawback is common in all solutions that split power based on two converters for designing VLC drivers (Figure 2).

### 3.3. Analysis of the Switching Harmonic Components of the Proposed TIBuck DC/DC Converter

Although V_2_ is considered constant during a switching cycle in the TIBuck DC/DC converter, it can have switching harmonic components (i.e., output voltage ripple), which come from the auxiliary Buck DC/DC converter. Figure 10 is a representation of the filtering process in both converters and the effect of their switching harmonics on the output voltage of the proposed TIBuck DC/DC converter. Both filters are depicted in Figure 8. The output voltage V_2_ is the result of the low-pass filtering process of the switching node voltage v_sw−bu_(t) at the auxiliary Buck DC/DC converter. Its second-order filter is defined by L_bu_ and C_bu_, setting the cut-off frequency, f_c−bu_. Its spectral representation, |v_sw−bu_(f)|^2^, has a DC component, as well as switching harmonics at multiples of the switching frequency of the auxiliary Buck DC/DC converter, f_bu_. The cut-off frequency f_c−bu_ is chosen according to the switching frequency f_bu_, attenuating the output switching harmonic components. Figure 9 shows that point; although the frequencies f_bu_, f_sig_ and f_sw_ are sufficiently separated, part of the attenuated switching frequency harmonics of the auxiliary Buck DC/DC converter arise in V_2_, and they can lie within the communication signal bandwidth. For the sake of simplicity, this possibility is represented by the k_th_ harmonic of f_bu_. To minimize this effect, f_c−bu_ must be selected low enough so that the switching frequency f_bu_, and therefore the ripple on V_2_, is negligible in the communication bandwidth. This is not really a problem, because the auxiliary Buck DC/DC converter does not need high bandwidth.

The spectral representation of the switching node voltage of the proposed TIBuck DC/DC converter, v_sw_(t), is |v_sw_(f)|^2^. It has components coming from the V_2_ ripple (as f_bu_ harmonics), the communication signal spectrum (centred in f_sig_) and the switching harmonic components of the switching frequency of the TIBuck DC/DC converter, f_ti_. Finally, the output filter of the TIBuck DC/DC converter is in charge of attenuating the f_ti_ switching harmonics and letting the communication signal spectrum pass. The result is depicted as |v_o_(f)|^2^. The cut-off frequency of the output filter of the TIBuck DC/DC converter, f_ti_, and the order of the filters are selected to achieve a certain bandwidth with low-level distortion in v_o_(t) (i.e., a low level of switching harmonic components in v_o_(t)). As analysed above, it is desirable to increase the bandwidth while reducing the switching frequency f_ti_. Equation (7) shows that f_c-ti_ must be selected in between the signal spectrum and the switching frequency. How close these frequencies can be is determined by the order of the output filter. The higher the order, the closer the frequencies can be, but the more difficult it is to design.

To summarize this section, the proposed solution of the VLC driver based on the split power technique using a proposed TIBuck DC/DC converter outperforms the Buck DC/DC converter in terms of efficiency and resolution for tracking the output voltage. Moreover, the modification of the TIBuck DC/DC converter with the auxiliary Buck DC/DC converter reduces the hardware complexity of previous approaches to achieve the split power technique. However, the price to pay is the loss in resolution for tracking the output voltage when the temperature of HB-LEDs increases.

## 4. Power Flow Analysis

As previously mentioned, the proposed TIBuck DC/DC converter adds an auxiliary Buck DC/DC converter to supply the voltage V_2_ from the input voltage V_1_. The overall architecture can be seen as two DC/DC converters which process different amounts of power at very different switching frequencies. Therefore, their individual efficiencies will affect the overall efficiency differently. A study of the efficiency of the TIBuck DC/DC converter was presented in [23] as a function of the voltages V_1_, V_2_ and its performance. Using this study as a baseline, this section presents a calculation of the overall efficiency of the proposed TIBuck DC/DC converter based on the efficiency of each individual converter.

### 4.1. Division of Output Power

The study starts by defining how much of the average output power undergoes a power conversion process for the TIBuck DC/DC converter alone (without considering the auxiliary buck converter). From Figure 11a, the average output power P_out_ is defined as
(17)$$P_{out}=V_{o}\cdot I_{o}=\frac{V_{o}\left(V_{o}-nV_{th}\left(T\right)\right)}{nR_{\Omega }}.$$
with R_Ω_ the dynamic resistance of HB-LED. If V_o_ is higher than the HB-LED load threshold voltage, nV_th_(T), then the equivalent of the HB-LED load can be used as in Figure 11b. Hence, the current driven by the HB-LED load is
(18)$$I_{o}=\frac{V_{o}-nV_{th}\left(T\right)}{nR_{\Omega }}.$$

To calculate the fraction of the output power that does and does not undergo the power conversion process, the output load is divided in such way that the voltage across the lower side is V_2_ and the voltage across the top side is V_o_−V_2_, as depicted in Figure 11c. For this equivalence to be true, the current driven by both parts of the HB-LED load must be I_o_, producing this equation for the top side
(19)$$\frac{V_{o}-V_{2}-nV_{th1}}{nR_{\Omega 1}}=\frac{V_{o}-nV_{th}\left(T\right)}{nR_{\Omega }},$$
and this equation for the bottom side
(20)$$\frac{V_{2}-nV_{th2}}{nR_{\Omega 2}}=\frac{V_{o}-nV_{th}\left(T\right)}{nR_{\Omega }}.$$

The values of the components of both equivalents are calculated from Equations (19) and (20), obtaining
(21)$$nR_{\Omega }=nR_{\Omega 1}=nR_{\Omega 2},\;\;nV_{th1}=nV_{th}\left(T\right)-V_{2},\;\;nV_{th1}=nV_{th}\left(T\right)-V_{2},$$

The expression of P_ti_ and P_fr_ is obtained from Figure 11 as
(22)$$P_{ti}=\left(V_{o}-V_{2}\right)\cdot I_{o},$$
(23)$$P_{fr}=V_{2}\cdot I_{o},$$
with P_ti_ being the power of the top side of the equivalent that undergoes power conversion and P_fr_ the power of the bottom side of the equivalent that does not undergo power conversion.

### 4.2. Input Power Calculation

The next step is to obtain the power delivered by the voltages V_1_ and V_2_ to the HB-LED load. It is important to point out that this power calculation is without taking the proposed TIBuck DC/DC converter into account, so V_1_ and V_2_ will be considered as individual voltages sources, as depicted in Figure 3. Later in this study, the generation of V_2_ from V_1_ and its respective power conversion will be considered. The power of each input voltage can be easily calculated from Figure 2.

In terms of average values, the average driven by the inductors of the high-order output filter of the TIBuck DC/DC converter is the same as the average current through the load Io. With D_ti_ being its duty cycle, the average currents of the input voltages I_v1_ and Iv2 can be easily calculated, as follows:(24)$$I_{v1}=D_{ti}\cdot I_{o},$$
(25)$$I_{v2}=\left(1-D_{ti}\right)\cdot I_{o}.$$

Hence, the power of each input voltage P_v1_ and P_v2_ can be obtained from Equations (24) and (25), as follows:(26)$$P_{v1}=V_{1}{\cdot D}_{ti}\cdot I_{o},$$
(27)$$P_{v2}=V_{2}\cdot \left(1-D_{ti}\right)\cdot I_{o}.$$

### 4.3. Power Flow of the TIBuck DC/DC Converter

The power flow is obtained by comparing the input power from Equations (26) and (27), and output power from Equations (22) and (23). The power flow diagram of the TIBuck DC/DC converter is shown in Figure 12. The power P_fr_ accounts for the input power P_v2_ and a fraction of P_v1_ called P_v1fr_. From Equations (23) and (27), P_v1fr_ can be calculated as
(28)$$P_{v1fr}=P_{fr}-P_{v2}=V_{2}{\cdot D}_{ti}\cdot I_{o},$$
with the power P_ti_ being the fraction of the output power that undergoes power conversion and P_v1fr_ the part that does not. The efficiency of the power conversion process in the TI Buck DC/DC converter is modelled by η_ti_, resulting in a power demanded from the input voltage V1 of
(29)$$P_{in-ti}=\frac{P_{ti}}{\eta_{ti}}+P_{v1fr}.$$

### 4.4. Power Flow of the Proposed TIBuck DC/DC Converter

The previous steps were made considering V_2_ as an input power, while in the proposed version this voltage is obtained from the input voltage V_1_. The complete power flow of the proposed TIBuck DC/DC converter, including the efficiency of the auxiliary Buck DC/DC converter, is shown in Figure 13. The output power of the system, P_out_, is the sum of P_ti_ and P_fr_. The new fraction of the power flow is the contribution of the generation of V_2_ from V_1_. The power P_in−bu_ is processed by the auxiliary Buck DC/DC converter with an efficiency of η_bu_ giving P_v2_, with
(30)$$P_{in-bu}=\frac{P_{v2}}{\eta_{bu}}.$$

Finally, the input power, Pin, can be computed as
(31)$${P_{in}=P_{in-ti}+P}_{in-bu}.$$

### 4.5. Overall Efficiency

The objective of this subsection is to calculate the efficiency of the proposed TIBuck DC/DC converter and to determine how that is affected by the different design parameters. With this efficiency being η_t_, the ratio between the output power, P_out_, and the input power, P_in_, by using Equations (29)–(31), η_t_ can be written as follows
(32)$$\eta_{t}=\frac{P_{out}}{P_{in}}=\frac{P_{out}}{\frac{P_{v2}}{\eta_{bu}}+\frac{P_{ti}}{\eta_{ti}}+P_{v1fr}}.$$

If Equations (17), (22), (27) and (28) are applied to Equation (32), it can be expressed in terms of the different design parameters:(33)$$\eta_{t}=\frac{V_{o}\cdot I_{o}}{\frac{V_{2}\cdot \left(1-D_{ti}\right)\cdot I_{o}}{\eta_{bu}}+\frac{\left(V_{o}-V_{2}\right)\cdot I_{o}}{\eta_{ti}}+V_{2}{\cdot D}_{ti}\cdot I_{o}}.$$

Equation (33) is the general approach of the efficiency, with no regard for the type of load or the type of application. Some of the values can be specified for a string of n HB-LEDs as a load in a VLC driver. The average output voltage value can be taken from Equation (2), giving
(34)$$V_{o}={nV}_{th}\left(T\right)+n\frac{V_{\Omega }}{2}.$$

As previously mentioned in Section 3 (Figure 9), the voltage feedback loop of the auxiliary Buck DC/DC converter changes the V_2_ value to correct for the temperature changes of the HB-LED string. By using Equations (10) and (34) in (33), the following expression for efficiency is obtained as a function of temperature with an HB-LED string as a load:(35)$$\eta_{t}=\frac{V_{th}\left(T\right)+\frac{V_{\Omega }}{2}}{\frac{V_{th}\left(T\right)}{\eta_{bu}}\left(1-D_{ti}\right)+\frac{V_{\Omega }}{\eta_{ti}}+V_{th}\left(T\right){\cdot D}_{ti}}.$$

The output current, I_o_, and the number of HB-LEDs has been simplified. Therefore Equation (35) does not depend on them. The only parameter left is the duty cycle of the TIBuck DC/DC converter, D_ti_. Finally, adding Equations (10) and (34) into Equation (8) and solving for D_ti_ leads to
(36)$$D_{ti}=\frac{nV_{\Omega }}{2\left(V_{1}-nV_{th}\left(T\right)\right)}.$$

### 4.6. Temperature and Partial-Efficiency Dependencies of the Overall Efficiency of the Proposed TIBuck DC/DC Converter

It can be seen from Equations (35) and (36) that η_t_ has a clear dependency on the efficiencies of the individual DC/DC converters (TIBuck and auxiliary Buck), as well as the HB-LED characteristics, some of which vary with temperature. To have a better idea about how the changes in these parameters affect the efficiency η_t_, a parametric analysis has been performed, sweeping through different values.

In the first analysis, only one of the efficiencies of the individual converters has been swept. The HB-LED parameters were taken from standard parts, assuming no variation in temperature (i.e., V_th_ = 3 V and V_Ω_ = 1 V). The value of V_1_ is obtained from Equation (9). Figure 14 shows how the overall efficiency of the proposed TIBuck DC/DC converter acting as a VLC driver, η_t_, varies with the variation in the efficiency of each individual DC/DC converter (i.e., with variations of η_ti_ and η_bu_). In each sweep, the efficiency of one DC/DC converter varies from 0.8 to 1, while keeping the other efficiency at 1. It is clear that the efficiency of the proposed TIBuck DC/DC converter depends more strongly on the value of η_bu_. The auxiliary Buck DC/DC converter processes higher power, making its efficiency much more important. This becomes an important advantage, because the auxiliary buck converter operates at a lower switching frequency, and it is easier to achieve higher efficiency here than in the TIBuck DC/DC converter, which operates at a higher frequency.

The same analysis of efficiencies of the individual DC/DC converters is shown in Figure 15, but at three different temperatures, with T_1_ > T_2_ > T_3_. Now the temperature effect over the HB-LEDs is analysed by considering different threshold voltages at different temperatures (V_th_(T_1_) = 2 V, V_th_(T_2_) = 2.5 V and V_th_(T_3_) = 3 V). The same trend can be seen when the temperature rises, but now the dependency of η_t_ with respect to η_bu_ becomes stronger. In the case of the dependency of η_t_ with respect to η_ti_, the change with the temperature is slighter.

To examine the effect of temperature in η_t_ more closely, the threshold voltage will be varied. Since there is little change with respect to η_ti_, this is kept constant at a realistic value for a high frequency converter with η_ti_ = 0.9. Then η_t_ is calculated by sweeping V_th_ from 2 V to 3 V at different values of η_bu_, ranging from 1 to 0.9. The results are shown in Figure 16. As expected from the previous analysis, there is always a falling tendency on η_t_ when the threshold voltage falls, but the lower the efficiency η_bu_, the more pronounced the fall.

In conclusion, the efficiency of the proposed TIBuck DC/DC converter is going to strongly depend on the efficiency of the auxiliary Buck DC/DC converter, operating at low switching frequency. It is important to note that the dependency on the efficiency of the TIBuck DC/DC converter is lower when this operates at high switching frequency. This result strengthens the previously mentioned design philosophy for the proposed TIBuck DC/DC converter based on splitting power: to focus the design of each individual converter on different goals. The design of the TIBuck DC/DC converter will be focused on communication performance, while the auxiliary Buck DC/DC converter will be focused on efficiency.

## 5. Experimental Results

To validate the proposed TIBuck DC/DC converter working as a VLC driver, a prototype was built to prove the concept, following the prior analysis and guidelines. Figure 17 shows the real prototype and Figure 18 shows the schematic and the block diagram of the control stage. The control of the TIBuck and the auxiliary Buck DC/DC converters were integrated in a FPGA Nexys A7. The FPGA implements the output-current feedback control loop by using the measured average output current, I_o_, and the lighting reference as inputs. The average output current is controlled by means of varying V_2_, following Equation (14). Moreover, the FPGA controls the TIBuck DC/DC converter, which operates in an open loop, generating the communication signal according to the modulator block and the input bits.

As general specifications, the TIBuck DC/DC converter supplies an HB-LED load of 8 PC-LED XLamp MX-3 in series and an input voltage V_1_ = 28 V. The input voltage was selected as being close to but lower than the maximum voltage of the HB-LED string, based on Equation (9). V_1_ was selected to be 12% lower, to compensate for the degradation of resolution for tracking the output voltage, due to the increase in temperature in the HB-LEDs previously shown in Figure 7. The value was selected based on the experimental behaviours of the HB-LED load. This allows us to partially counteract the temperature effect on the resolution by designing the voltage V_1_ closer to the output voltage of the HB-LED load at room temperature. This specification can be easily met, because the maximum voltage of each HB-LED was expected to drop from 4 V to 3.5 V when the working was stabilized. The average current, I_o_, was kept at 0.25A (the middle of the HB-LED linear range) and controlled by the FPGA. The next subsections further explain the design process.

### 5.1. Design of the Auxiliary Buck DC/DC Converter

The auxiliary Buck DC/DC converter was designed with a switching frequency, f_bu_, of 100 kHz, and a second-order output filter with a cut-off frequency, f_c−bu_, of 10 kHz, one decade below the switching frequency. As shown in Figure 10, there is a trade-off on the selection of f_bu_ and f_c−bu._ The higher the switching frequency, the smaller and lighter the converter, which is an advantage in VLC drivers. On the other hand, the higher the switching frequency, the higher the power losses and the more difficult it is to design the filter. In this design, a conservative approach was chosen, making the output voltage ripple negligible.

The auxiliary Buck DC/DC converter works in a closed loop, and the control was implemented in the FPGA. The output current i_o_(t) of the HB-LED load is measured by a shunt resistance in series. The current i_o_(t) has both DC and communication-signal components. Since only the average value, I_o_, needs to be controlled, there is a low-pass filter, so the only component sampled by the FPGA ADC is Io. The PI compensator was designed with 10 Hz of bandwidth.

By means of this control, the auxiliary Buck DC/DC converter ensures the HB-LED always works in the middle of its linear region, regardless of temperature. The slow nature of thermal behaviour simplifies the design of the current control, which does not need a fast response or a wide bandwidth. It is worth noting that the dynamics of the output average control throughout V_2_ in the proposed TIBuck DC/DC converter can be approximated by the dynamic of the auxiliary Buck DC/DC converter if the following conditions are met: f_bu_ << f_ti_ and f_c−bu_ << f_c-ti_. These conditions are implicit in Figure 10. If they are met, Equation (14) holds true. This means that only f_bu_ and the output filter of the auxiliary Buck DC/DC converter limit its dynamics.

The list of components used for the auxiliary Buck DC/DC converter can be found in Table 1. The converter was implemented in a synchronous configuration (to increase the overall efficiency of the proposed TIBuck DC/DC converter, following the conclusion in Section 4) by two MOSFETs integrated in the same chip CSD88539 (switches Q_bu−1_ and Q_bu−2_) and a half-bridge gate driver ISL6700.

### 5.2. Modulation Scheme

To test the communication capability of the proposed TIBuck DC/DC converter as a VLC driver, a 64-QAM modulation scheme was used. The communication signal modulates the amplitude and phase and, due to its high complexity and wide use in VLC [3,4,5], it allows proof of concept of the communication capability of the prototype. The carrier frequency must lie within the HB-LED bandwidth, hence a carrier frequency, f_sig_, of 1 MHz was used. By using a symbol period, T_sym_, of 4 carrier periods, the maximum bit rate achieved was 1.5 Mbps.

### 5.3. Design of the TIBuck DC/DC Converter

The TIBuck DC/DC converter must be able to generate a 1 MHz communication signal with low distortion and high efficiency. Its switching frequency, f_sw_, the order and the cutoff frequency of the filter, f_c_, were designed according to the modulation scheme chosen previously. There is also a trade-off between efficiency, resolution achieved for tracking the output voltage and filter design. The lower the switching frequency, the higher the efficiency and the resolution. Moreover, there is a limit to how much the switching frequency can be reduced. As shown in Equation (7), a condition must be fulfilled regarding f_ti_, f_c-ti_ and f_sig-max_ (which is f_ws_ = f_ti_ and f_c_ = f_c_ − t_i_), allowing the filter to separate the signal spectrum from the switching harmonic components of the output voltage. Studies and guidelines about how close these frequencies can be placed were performed and reported in [18,21]. The switching frequency, f_ti_, was selected at 10 MHz, a decade above the carrier frequency. Following the aforementioned guidelines, the filter used was a 6th Butterworth filter with a cut-off frequency, f_c-ti_, of 2.5 MHz. The reactive elements for the output filter are shown in Table 2. The converter was implemented in an asynchronous configuration using the high frequency RF MOSFETs PD84010S-E for Q_ti_ and the fast diode UPS115UE3 for S_ti_. Due to its high switching frequency, a high-speed gate driver, EL7155CSZ, was used. Since Q_ti_ is not referred to the circuit ground, a fast signal isolator, ISO721, is needed between the FPGA and the gate driver.

### 5.4. Experimental Waveforms

The prototype was experimentally evaluated as a VLC driver to prove the concept, generating a communication signal following the 64-QAM modulation scheme. During the tests, the efficiency of the converter was measured, and the most significant waveforms were obtained to illustrate communication performance.

The efficiency calculation involves the measurement of the both the input and output power. The input power was directly measured by high-precision multimeters, measuring input voltage and input current provided by V_1_. The output power needed to be measured using the oscilloscope because of the high frequency of the communication signal. Both the output voltage and the output current were measured, stored and post-processed to calculate the output power. The output voltage was directly measured by an oscilloscope probe and the output current was measured using a high-precision low-inductance shunt resistor with high thermal stability. The high thermal stability in the shunt is mandatory, since it is connected in series and close to the HB-LED load. The low stray inductance and the proximity of the shunt are also required, due to the nature of the high frequency of f_sig_. During regular operation, when maximum communication power was processed (the worst case, because the amplitude of the communication signal could be at a maximum in certain symbols), the efficiency achieved was 94%, with the output power being 7 W. Figure 19 shows the most significant waveforms from the converter during its operation as a VLC driver (i.e., v_o_(t) and i_o_(t)). The average value of the output current, I_o_, was kept at 0.25 A by the action of the output voltage loop of the auxiliary Buck DC/DC converter. The instantaneous value of v_o_(t) (with a small portion of DC) was controlled by the TIBuck DC/DC converter according to v_sig_(t). The output current was never 0 A.

Therefore, the HB-LEDs do not work outside their linear regions. The voltage v_sw_(t) is the switching node voltage of the TIBuck DC/DC converter and it is a square waveform modulated in PWM that varies between V_1_ and V_2_, as was expected. The top value V_1_ is constant, while V_2_ changes according to the current loop control. At room temperature, the average voltage of this HB-LED load would be 28 V (data extracted from the datasheet). According to Figure 19, the average voltage V_o_ was 24 V, which is 14% lower. This value is on the same page as the correction made at the beginning of the design when selecting the voltage V_1_. This allows partial counteracting of the temperature effect on the resolution by designing the voltage V_1_ closer to the operation value rather than at room temperature.

To validate the correct conversion of the communication signal into variations of light, the emitted light was measured. A high-speed and wide-bandwidth optical receiver Thorlabs PDA10A-E was placed at 0.4 m in front of the HB-LED load. The voltage v_rx_(t) is the output of the receiver, which was proportional to the intensity of light, checking that the HB-LED load is fast enough and its bandwidth was wide enough to allocate the communication scheme. Figure 19 shows a 12-symbol transmission of the 64-QAM modulation scheme. The different phases and amplitudes of the modulation symbols were correctly reproduced.

Following that, the dynamic behaviour of the prototype was evaluated by performing amplitude and phase changes. As Figure 20 shows, the designed VLC driver required a settling time of less than one communication-signal period to perfectly track the reference after the change. This settling time is short enough to address the symbol selected (T_sym_ = 4/f_sig_) to reach the 1.5 Mbps bit rate.

### 5.5. Communication Performance

One method to evaluate whether the symbols are well reproduced or not is by calculating the error vector e_v_ [26], which is a metric of communication performance. It is defined as
(37)$$e_{v}^{\left[i\right]}=v_{rx}^{\left[i\right]}-v_{id}^{\left[i\right]},$$
where $v_{rx}^{\left[i\right]}$v is the i^th^ symbol received and $v_{id}^{\left[i\right]}$ is the ith sent symbol. These symbols correspond to an ideal sent symbol with a certain amplitude and phase. The error vector calculates the distance between the received and sent symbol. The bigger the difference between symbols, the longer the distance and the bigger the error vector. This test is normally performed over a long random sequence, in this case up to 256 symbols. For a sequence of m symbols, the Error Vector Magnitude (EVM_rms_) is obtained, which is the normalized root mean square of the error vector e_v_ over a sequence of m symbols. It is calculated as
(38)$${EVM}_{rms}\left[\%\right]=100\cdot \sqrt{\frac{\sum_{i=1}^{m}{\left|v_{rx}^{\left[i\right]}-v_{id}^{\left[i\right]}\right|}^{2}}{\sum_{i=1}^{m}{\left|v_{id}^{\left[i\right]}\right|}^{2}}}.$$

The value of EVM_rms_ is given as a percentage, measuring the error over a whole sequence. The lower the value, the better the communication performance. The prototype reached an error of 12% at a distance of 0.4 m.

Other tests could have been performed, such as Bit Error Rate (BER) or the error according to the reception distance, but the performance of these tests depends on the demodulator as well as the optical receiver, which must be designed for this specific application. The design of the demodulator and the optical receiver are outside the scope of this paper.

### 5.6. Comparison with Other Approaches

Table 3 shows a comparison with the state-of-the-art power-efficient VLC drivers based on PWM DC/DC converters able to reproduce advanced modulation schemes.

The proposed TIBuck DC/DC converter outperforms traditional Buck DC/DC converter solutions in terms of efficiency, bit rate and moderate EVM_rms_., even operating at higher switching frequency. Moreover, the hardware complexity of the proposed solution is similar to that of the Buck DC/DC solutions presented to design high-bandwidth VLC drivers (i.e., two-phase structures with high-order filter).

The proposed TIBuck DC/DC converter’s natural way to split power simplifies previous approaches using this technique, because it does not need two input voltage sources (one of which is isolated) or one isolated DC/DC converter. In addition to simplicity, high communication capabilities and high efficiency are maintained, with good design of the auxiliary Buck DC/DC converter.

It is important to note that Table 3 highlights the ratio between the maximum amplitude and the average value of i_o_(t) (i.e., i_omax_/2I_o_), which gives us an approach to assess of the ratio between communication power and lighting power. The TIBuck DC/DC converter maximizes communication power compared to other approaches, which minimize the amplitude of the communication to reach high efficiency (e.g., [27]).

## 6. Conclusions

The Buck DC/DC converter is a good approach for the last step of the power supply chain as a VLC driver. Some authors have proposed multi-phase solutions with high-order filters to achieve both high bandwidth and high efficiency. This is the context in which the technique of splitting the power between two DC/DC converters is proposed to further improve both the resolution of output voltage tracking and the efficiency of the VLC driver. To that end, one converter is in charge of communication tasks and the other handles lighting tasks. However, the real implementation of this solution introduces hardware complexity due to isolated issues either in previous steps of the power supply chain or in the VLC driver itself.

This work introduces a proposed version of the TIBuck DC/DC converters, both to outperform solutions based on Buck DC/DC converters and to simplify the hardware complexity of previous solutions that use the split power technique. By naturally splitting power and with the help of the auxiliary Buck DC/DC converter, high efficiency, high resolution in output voltage tracking, and great communication capabilities are achieved as a VLC driver. All improvements are based on a deep analysis of the proposed TIBuck DC/DC converter as a VLC driver in terms of interaction with the proposed Buck DC/DC converter, evaluation of switching harmonic components, filtering effort, the temperature dependency of the resolution and the nature of the power flow.

Finally, in other to prove the concept, a 7 W output power experimental prototype was built with 94% efficiency, reproducing a 64-QAM digital modulation scheme and achieving a bit rate of 1.5 Mbps, with an error in communication of 12%.

## Figures and Tables

**Figure 1 sensors-24-06392-f001:**
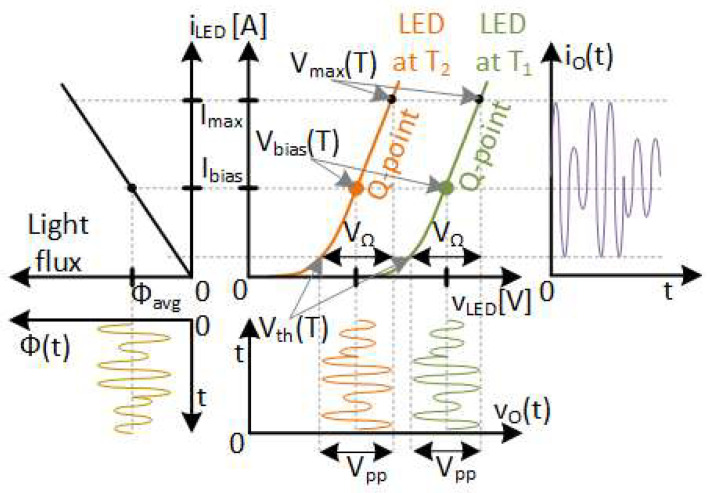
Temperature effects on the I-V-Flux characteristic of the HB-LED.

**Figure 2 sensors-24-06392-f002:**
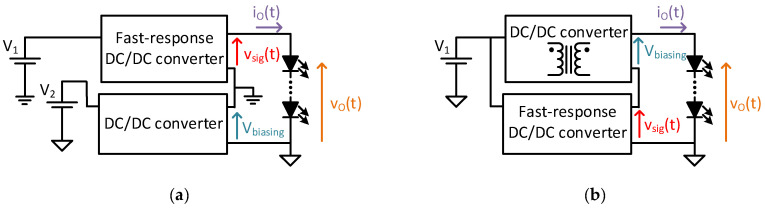
Two implementation examples of the split power technique: (**a**) with two isolated input voltage sources, one of them isolated (in [22]). (**b**) With an isolated DC/DC converter in the VLC driver structure.

**Figure 3 sensors-24-06392-f003:**
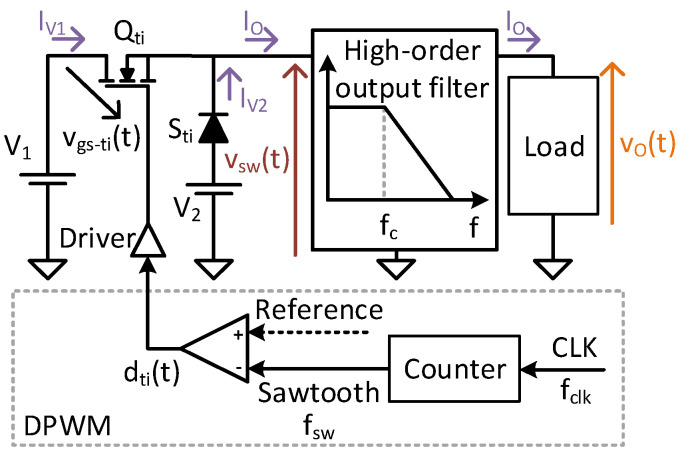
The TIBuck DC/DC converter topology and its control.

**Figure 4 sensors-24-06392-f004:**
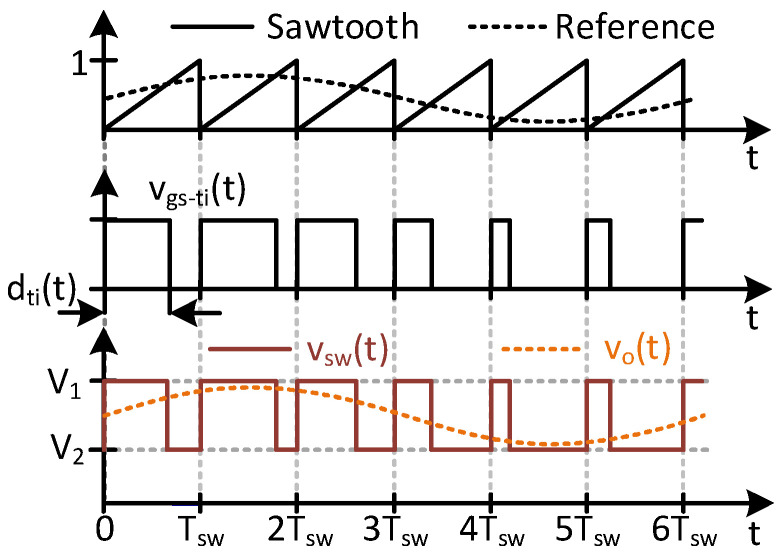
Most significant control waveforms of the TIBuck DC/DC converter.

**Figure 5 sensors-24-06392-f005:**
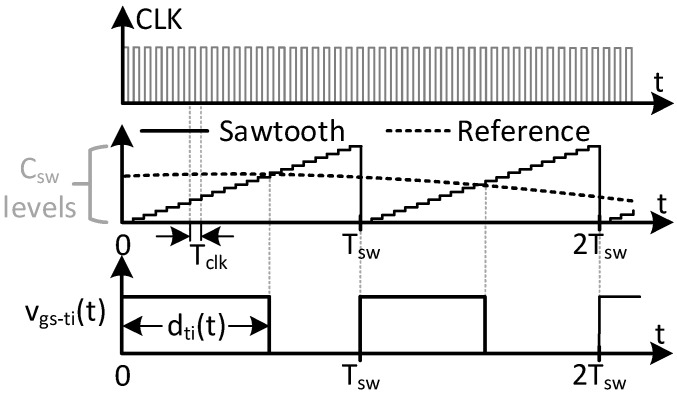
DPWM modulator and digital implementation of the sawtooth signal for two switching periods of Figure 4.

**Figure 6 sensors-24-06392-f006:**
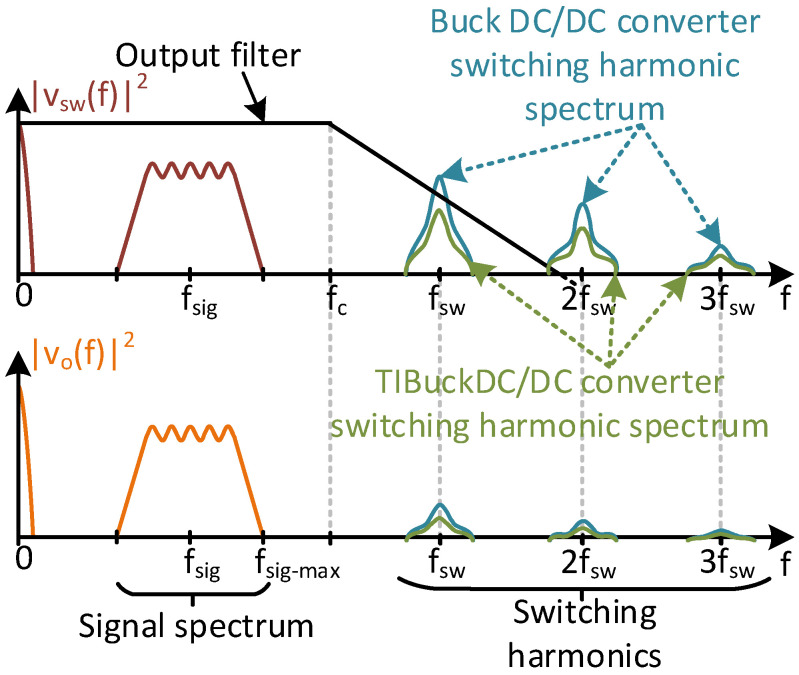
Spectral representation of the comparison between the Buck and TIBuck DC/DC converters in regards to filtering action of the output voltage.

**Figure 7 sensors-24-06392-f007:**
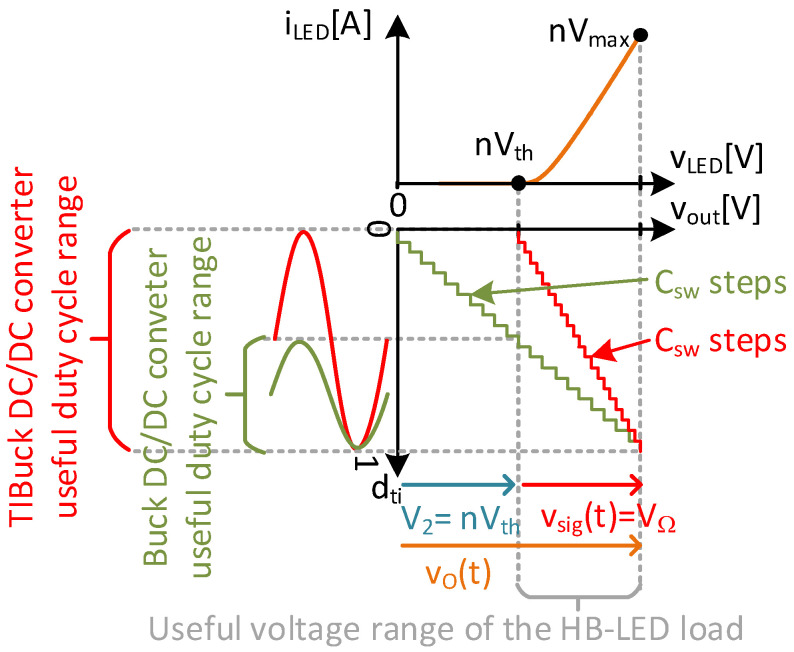
Representation of the output voltage range of the Buck and TIBuck DC/DC converters versus the useful voltage range of the HB-LED load and their impact on resolution to track the output voltage.

**Figure 8 sensors-24-06392-f008:**
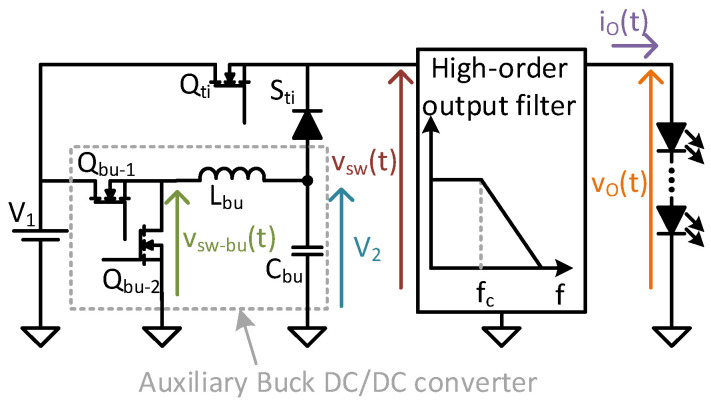
The proposed TIBuck DC/DC converter with only one input voltage source, V_1_, provided by the auxiliary Buck DC/DC converter.

**Figure 9 sensors-24-06392-f009:**
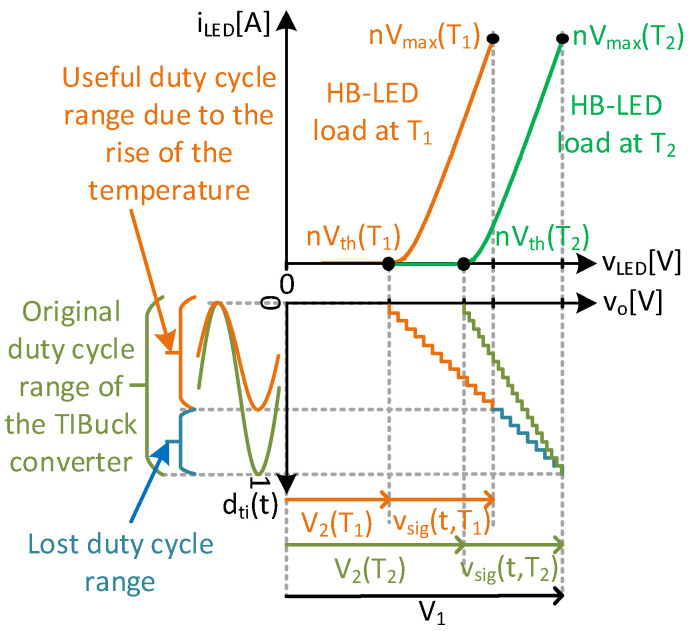
Representation of the output voltage range of the TIBuck DC/DC converter versus the useful voltage range of the HB-LED load and the impact on the resolution for tracking the output voltage when temperature changes occur.

**Figure 10 sensors-24-06392-f010:**
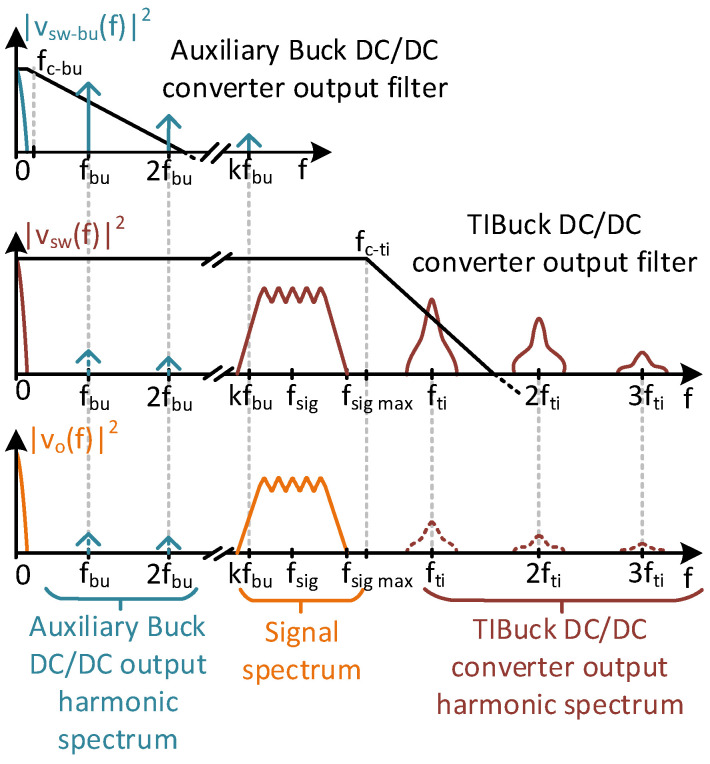
Spectral representation of the interaction between the TIBuck and auxiliary Buck DC/DC converter with regard to filtering action of the output voltage.

**Figure 11 sensors-24-06392-f011:**
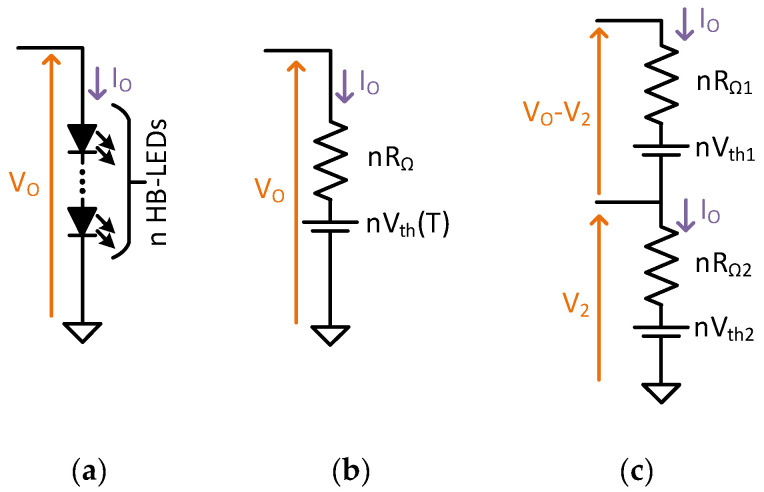
Division process of the output HB-LED load (only one string for the sake of simplicity). (**a**) Initial scenario. (**b**) Modelling the HB-LED load. (**c**) Dividing the HB-LED load in two parts.

**Figure 12 sensors-24-06392-f012:**
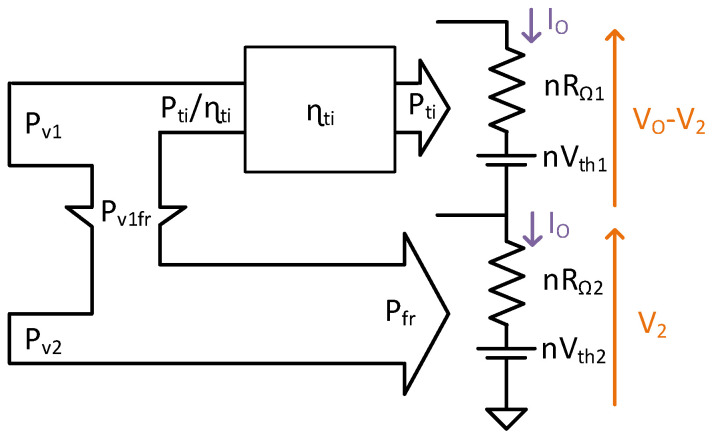
Power flow diagram of the TIBuck DC/DC converter.

**Figure 13 sensors-24-06392-f013:**
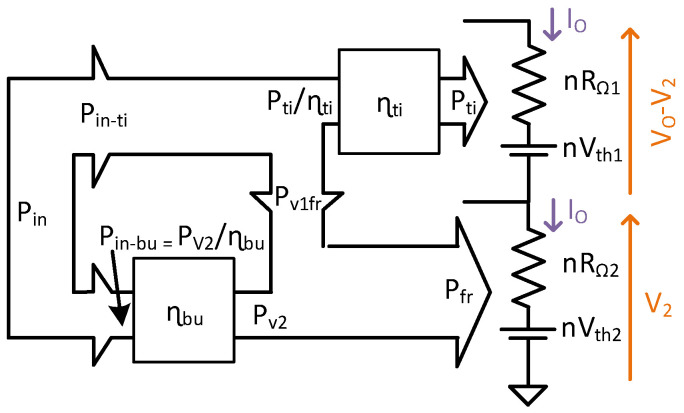
Power flow diagram of the proposed TIBuck DC/DC converter.

**Figure 14 sensors-24-06392-f014:**
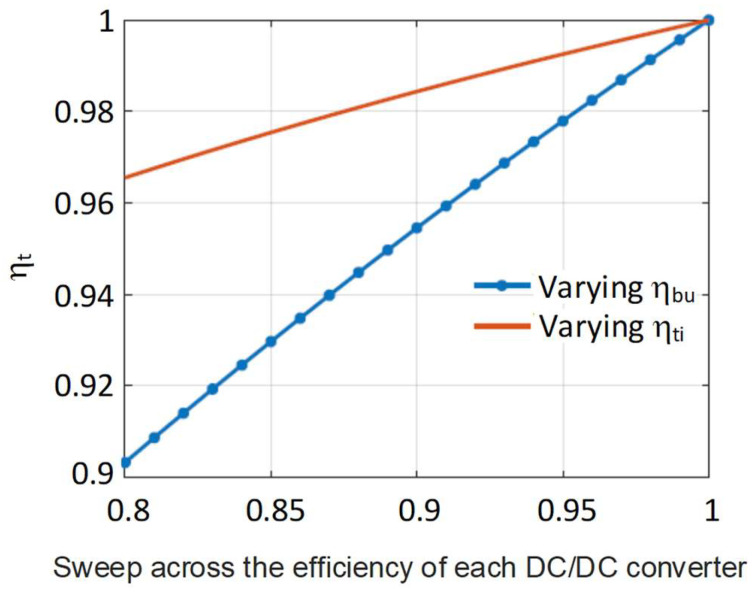
Effect of the variation in the efficiency of each individual DC/DC converter in η_t_.

**Figure 15 sensors-24-06392-f015:**
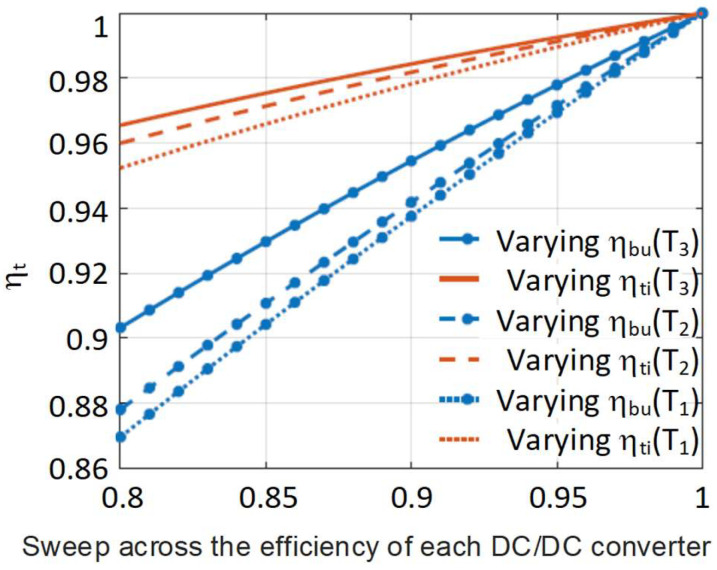
Effect of the variation of the efficiency of each individual converter in η_t_ at different temperatures.

**Figure 16 sensors-24-06392-f016:**
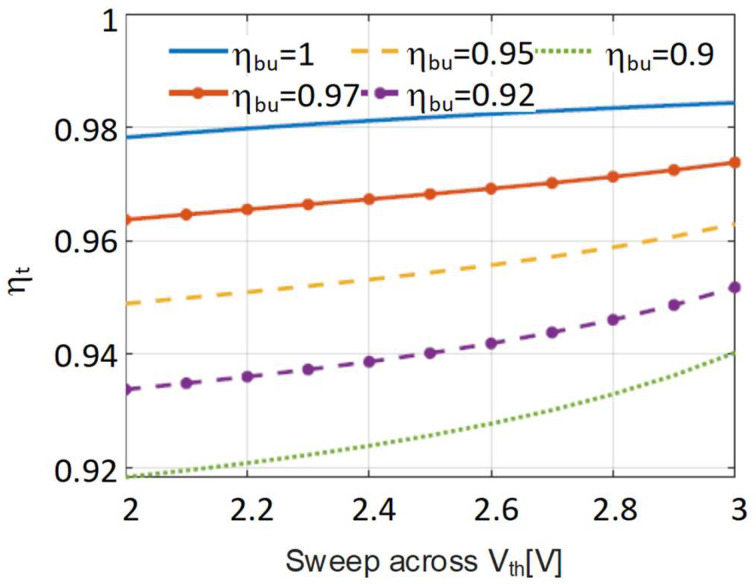
Effect of the variation of the threshold voltage of the LED with the temperature in η_t_ at different η_bu_.

**Figure 17 sensors-24-06392-f017:**
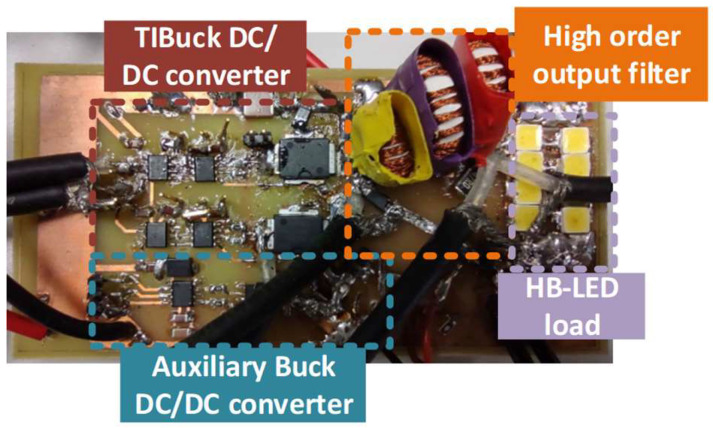
Photograph of the prototype of the proposed TIBuck DC/DC converter working as a VLC driver.

**Figure 18 sensors-24-06392-f018:**
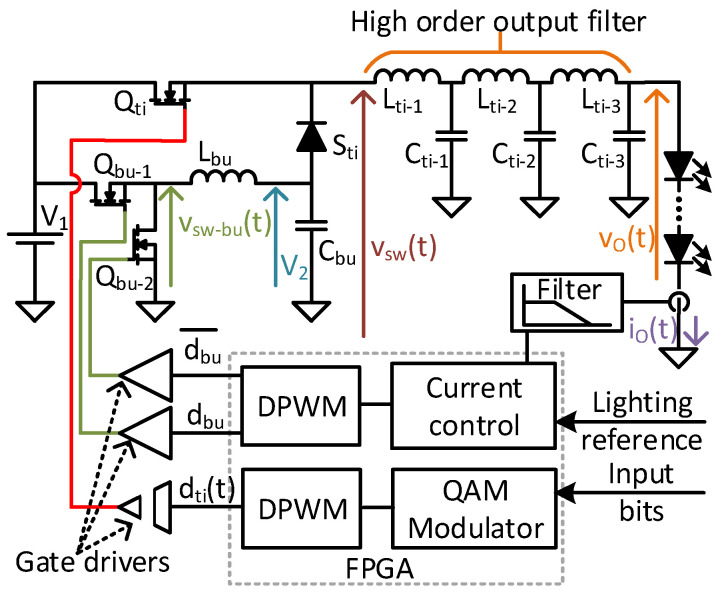
Schematic and control block diagram of the proposed TIBuck DC/DC converter prototype used as a VLC driver.

**Figure 19 sensors-24-06392-f019:**
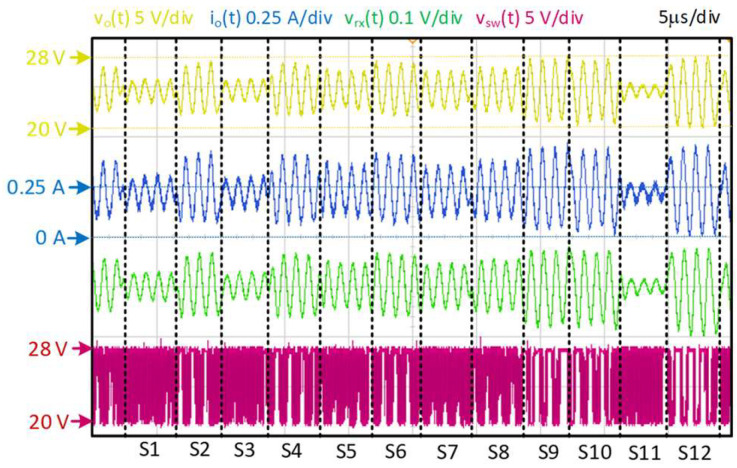
Most significant waveforms of the prototype acting as a VLC driver: output voltage, v_o_(t), output current, io(t), received light, v_rx_(t) and switching node voltage, v_sw_(t).

**Figure 20 sensors-24-06392-f020:**
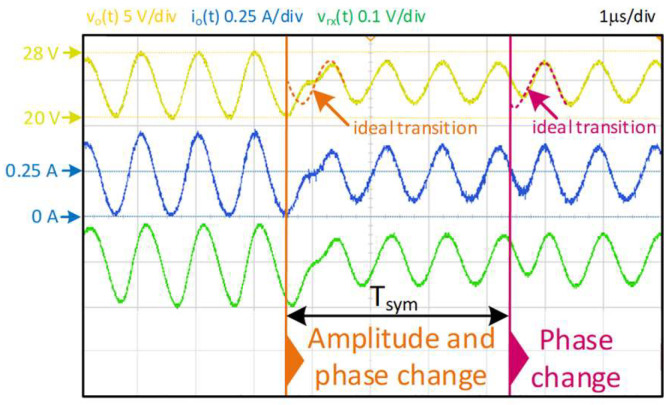
Detail of some waveforms when amplitude and phase changes occur. Output voltage, v_o_(t), output current, i_o_(t) and received light, v_rx_(t).

**Table 1 sensors-24-06392-t001:** List of components of the auxiliary Buck DC/DC converter.

Q_bu-1_ and Q_bu-2_	Gate Driver	L_bu_	C_bu_
CSD88539	ISL6700	49 μH	9 μF

**Table 2 sensors-24-06392-t002:** List of components of the auxiliary TIBuck DC/DC converter.

L_ti-1_	C_ti-1_	L_ti-2_	C_ti-2_	L_ti-3_	C_ti-3_
1.7 μH	9.9 nF	2.2 μH	9.9 nF	1.9 μH	5.72 nF

**Table 3 sensors-24-06392-t003:** Comparison between power-efficient VLC drivers based on conventional PWM DC/DC converters.

Ref.	Topology	Input Voltage Sources	Modulation Scheme	P_O_ (W)	i_omax_/2I_o_	f_S_ (MHz)	η(%)	Distance (cm)	Bit Rate (Mbps)	EVM_rms_ (%)
[21]	Two-phase Buck DC/DC converter with high-order filter	1	16-QAM	10	0.9	4.5	87.5	100	0.5	-
[22]	Two-phase Buck DC/DC converter with high-order filter + synchronous Buck DC/DC converter	2 ^1*^	OFDM	10	0.8	10 ^2*^	93.6	20	7.5	15
[27]	Fly-Buck DC/DC converter + Class B power amplifier	1	OFDM	20	0.42	0.1	94	50	6	5
This work	Proposed TIBuck DC/DC converter	1	64-QAM	7	1	10 ^3*^	94	40	1.5	12

^1*^ One of them isolated. ^2*^ This is the switching frequency of the DC/DC converter in charge of reproducing the communication signal. ^3*^ This is the switching frequency of the DC/DC in charge of both biasing and supplying the class B power amplifier.

## Data Availability

Data are contained within the article.

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
