# Peer review of "Performance Evaluation of the Two-Input Buck Converter as a Visible Light Communication High-Brightness LED Driver Based on Split Power"

_sensors, 2024, doi:10.3390/s24196392_

Round 1

Reviewer 1 Report

Comments and Suggestions for Authors

The article presents a novel, high-efficiency HB-LED driver for Visible Light Communication (VLC) that employs a Two-Input Buck (TIBuck) DC/DC converter. This design enhances efficiency and simplifies the power-splitting technique used in VLC drivers, avoiding the need for isolated power sources. The prototype demonstrated 94% efficiency, supporting a 64-QAM modulation scheme at a 1.5 Mbps bit rate with a 12% communication error. The following comments may further improve the quality of the paper.

1. Providing a constellation diagram for 64QAM communication will better demonstrate the low error rate of communication.

2. What is the maximum frequency of signal transmission supported by this circuit?

3. Will this driver introduce additional signal distortion compared to traditional drivers in the frequency domain? Is the signal input to the driver linear compared to the signal driving the LED?

Reviewer 2 Report

Comments and Suggestions for Authors

1. Fig. 2b: Is there a missing ground at one of the outputs of the DC/DC converters?

2. What are the parameters that limit the maximum bandwidth to 5MHz? How can you increase the bandwidth to achieve higher data rate?

3. The prototype of the proposed TIBuck DC/DC converter shown in Fig. 17 working as a VLC 555 driver is a bit bulky. How can you miniaturize it? Does miniaturization improve its performance? Would there be a tradeoff between miniaturization and thermal dissipation?

4. How do you address the thermal issue associated with the proposed DC/DC converter design?

5. What is the minimum power supply voltage for the proposed design to function properly? Would this minimum power supply voltage be higher or lower than existing designs in references [22, 23, 31]?

6. Table 3: What would be the data rate of the proposed TI Buck DC/DC converter if the distance is increased from 40cm to 100cm?

Round 2

Reviewer 1 Report

Comments and Suggestions for Authors

All the issues have been addressed. I therefore suggest accept this manuscript.